# Dirichlet latent modelling enables effective learning and sampling of the functional protein design space

Evgenii Lobzaev [1,2] & Giovanni Stracquadanio [1] ✉

Engineering proteins with desired functions and biochemical properties is pivotal for biotechnology and drug discovery. While computational methods based on evolutionary information are reducing the experimental burden by designing targeted libraries of functional variants, they still have a low success rate when the desired protein has few or very remote homologous sequences. Here we propose an autoregressive model, called Temporal Dirichlet Variational Autoencoder (TDVAE), which exploits the mathematical properties of the Dirichlet distribution and temporal convolution to efficiently learn high-order information from a functionally related, possibly remotely similar, set of sequences. TDVAE is highly accurate in predicting the effects of amino acid mutations, while being significantly 90% smaller than the other state-of-the-art models. We then use TDVAE to design variants of the human alpha galactosidase enzymes as potential treatment for Fabry disease. Our model builds a library of diverse variants which retain sequence, biochemical and structural properties of the wildtype protein, suggesting they could be suitable for enzyme replacement therapy. Taken together, our results show the importance of accurate sequence modelling and the potential of autoregressive models as protein engineering and analysis tools.

Recent advances in DNA synthesis and sequencing technologies coupled with high-throughput, automated experimental screening platforms are enabling the engineering of proteins with desired function and properties suitable to address biotechnology and biomedical challenges[1].

Nonetheless, the protein design space is exponentially large, with most regions harbouring non-functional biomolecules. Therefore, there has been a strong interest in developing methods, both experimental and computational, to explore the neighbourhood of known functional proteins to identify new variants likely to be functional with the aim of reducing the burden of downstream experimental validation. Experimental methods are usually based on the Directed Evolution (DE) framework[2], a process of mutation and selection that allows to explore variants of a known protein in an unbiased fashion. However, in a DE campaign, only a fraction of the variants are functional,

which makes the whole process expensive and difficult to scale for large molecules.

The development of biophysical models of protein folding, instead, has boosted the use of computational approaches to rationally design proteins and variant libraries[3]. However, these models can only approximate the physical principles underpinning protein folding and thus are limited in designing proteins characterised by complex properties, like flexibility, which are not directly related to their thermodynamic properties.

High-order information underlying complex protein features can now be learned by leveraging petascale information available for known protein sequences in public databases[4]. Machine Learning (ML) and, more recently, Deep Learning (DL), have been able to exploit this information and are propelling a paradigm shift in protein engineering. While initial ML models required experimental data to identify beneficial

[1]School of Biological Sciences, The University of Edinburgh, Edinburgh, United Kingdom. [2]School of Informatics, The University of Edinburgh, Edinburgh, United Kingdom. ✉e-mail: giovanni.stracquadanio@ed.ac.uk

mutations, recent auto-regressive models, like Variational Autoencoder (VAE)[5,6], have achieved state-of-the-art performances in mutation effect prediction[7–9] and variant library design tasks[10–12] using only sequence information. VAEs achieve these results by mapping known protein families or homologues into parametric probabilistic distributions, which can then be used to either design new proteins or evaluate the likelihood of a variant belonging to the input sequence dataset, hence retaining their characteristic features. From a biotechnology point-of-view, VAEs have the potential to generate large libraries of functional variants to screen, compared to randomised experimental approaches, like directed evolution, where most variants are non-functional.

Interestingly, while there have been a lot of efforts in developing efficient architectures to learn distribution parameters and design new sequences, protein families have always been modelled with a standard multivariate Gaussian distribution and using complex encoder and decoder architectures, which make them difficult to use in practice. However, decades of research in homology search have shown that the Dirichlet distribution is significantly more powerful at modelling amino acid frequencies and relationships and in finding remote homologous sequences[13], suggesting that using this distribution could substantially improve current auto-regressive model performances in generating a large library of diverse protein variants.

Here, we hypothesised that modelling protein families using a Dirichlet distribution and implementing the decoding and encoding layers using a Temporal Convolutional Network (TCN) will lead to an efficient auto-regressive model to (i) identify substantially different sequences with similar biochemical and structural features to those of a known wildtype protein and (ii) to capture biological and fitness constraints while being computationally tractable. To prove that, we developed a model, called Temporal Dirichlet Variational Autoencoder (TDVAE), which maps protein homologues on a Dirichlet distribution and uses Temporal Convolutional Networks (TCNs)[14] to learn distribution parameters and sample new protein sequences from it.

We then assessed the performances of TDVAE as a mutation effect prediction tool on an extensive dataset of mutagenesis experiments and showed that it achieves comparable state-of-the-art results while being 90% smaller than the current best auto-regressive model. We also performed an extensive hyperparameter analysis to show the robustness of our model to parameter settings and its superior performance to the same architecture when modelling the latent space as a canonical multivariate Gaussian distribution. Finally, we used TDVAE to design a library of human $\alpha$-galactosidase (AGAL) variants, a complex lysosomal enzyme whose inactivation causes Fabry disease and with a well-characterised mutational landscape[15]. Our results show that TDVAE generates a diverse library of variants while retaining the biochemical and structural properties of the human enzyme and avoiding pathogenic mutations. Moreover, TDVAE identifies mutational hotspots associated with improved enzymatic activity and biochemical properties, while not requiring any experimental information.

Taken together, TDVAE provides a new effective and efficient platform to design libraries of functional proteins using only sequence information.

## Results

We assessed the performances of our approach by using TDVAE as a mutation effect prediction and as a protein engineering tool, comparing and contrasting experimental results with state-of-the-art methods and experimental data. Here, we hypothesised that a model able to predict mutation effect should be sufficiently powered to design functional protein variants by learning sequence features associated with known functional homologous sequences.

### TDVAE performance in predicting protein mutation effects
We first assessed TDVAE performances as a mutation effect prediction model using 19 widely used mutagenesis datasets[16]. Each dataset

consists of a library of experimentally characterised variants of a given wildtype protein, with each variant annotated with a fitness score quantifying different phenotypes, ranging from enzyme activity to cell viability, normalised and log transformed such that the wildtype fitness is 0. Out of 19 mutational sets, 16 contain single-point mutations, whereas the remaining 3 contain mutants with multiple mutations with respect to the wild type. The input information provided to a model is a multiple sequence alignment (MSA) generated as part of the EVMUTATION pipeline[7], which contains homologues of the wildtype sequence. Given a mutagenesis dataset and a model which outputs an effect score measuring the functional impact of one or more mutations, the model performances are then reported in terms of Spearman's correlation between the experimental fitness scores and the predicted mutation effect scores.

A recent comparative analysis of state-of-the-art models for mutation effect prediction showed that the unsupervised probabilistic model DeepSequence[8] consistently reported the best performance across all proteins in our dataset. DEEPSEQUENCE outputs a mutation effect score for a given variant sequence as the ratio $S_m = \log[p(\boldsymbol{x}_{mut}|\boldsymbol{\theta})/p(\boldsymbol{x}_{wt}|\boldsymbol{\theta})]$, where $p$ is replaced by the Evidence Lower Bound (ELBO); in practice, $S_m$ represents the log-likelihood of a variant $\boldsymbol{x}_{mut}$ relative to the wildtype sequence. This score has been shown to be predictive of mutation effects and can be learned without fitting the model to experimental data[7].

Here, we compared TDVAE performances against DEEPSEQUENCE, using the available PYTORCH implementation; specifically, we used the proposed Bayesian decoder and the default parameters as reported in the original study[8,9]. To perform a fair comparison and quantify the contribution of our Dirichlet latent space modelling, we adapted TDVAE to use the same sparse one-hot encoding layer as in DEEPSEQUENCE, to avoid any representational bias. We then used a single block of stacked dilated causal 1D convolutional layers with kernel size 3, intermediate channel size of 128, 20% dropout for regularisation, and a 50-dimensional latent space, with a symmetric Dirichlet distribution with $\alpha = 1.0$ as prior, whereas the number of causal convolutional layers is computed according to Eqn. (3). Unlike DEEPSEQUENCE, we did not use Variational Inference (VI) on the decoder weights or any structured parametrization in the final layer but again relied on a single block dilated convolutional layer[10]. For each dataset, we trained each model 5 times with 5 different random seeds using a mini-batch of 256 sequences and then computed 2, 000 ELBO samples per mutant to estimate model-specific mutation effect scores to be correlated with experimental data. Since these models are usually computationally taxing to train, we also tested a version of TDVAE, dubbed LOW MEM, which uses a mini-batch of only 4 sequences, as an alternative that can be readily adopted in consumer GPU hardware.

Experimental results showed that, in 17 out of 19 datasets, TDVAE performed better than DEEPSEQUENCE with up to 6% increase in Spearman's correlation, as for the *POLG* dataset (see Fig. 1A); specifically, the best performance was obtained by TDVAE on 8 datasets and by TDVAE LOW MEM on 9 datasets. When comparing the robustness of models' performance, that is, the best performance on average across 5 independent runs, performances were comparable; specifically, TDVAE achieved the best average performance on 7 datasets, TDVAE LOW MEM on 6 datasets and DeepSequence on the remaining 6 datasets. It is important to note that differences in performances are limited, albeit improvements of up to 5% in Spearman's correlation were found depending on the protein, e.g., *DLG4*. Interestingly, out of the 19 datasets, the LOW MEM version generally performed poorly on only 3 datasets, i.e., the difference in correlation is more than 10% compared to the best model, namely *GAL4*, *POLG* and *PABP*, suggesting that even a less optimal training process can still produce satisfactory results and can also be beneficial as smaller batches might act as regularizer during training (see Supplementary Data 1).

**A**

Model  DeepSequence  TDVAE  TDVAE (Low mem)

(Figure 1A: Box plots of Spearman's correlation across protein datasets: BG_STRSQ (Abate, 2015), BLAT_ECOLI (Ostermeier, 2014), BLAT_ECOLI (Palzkill, 2012), BLAT_ECOLI (Ranganathan, 2015), BLAT_ECOLI (Tenaillon, 2013), BRCA1_E3 (Fields, 2015), BRCA1_Y2H (Fields, 2015), DLG4_RAT (Ranganathan, 2012), GAL4_YEAST (Shendure, 2015), GFP (Sarkisyan, 2016), HSP82_YEAST (Bolon, 2016), KKA2_KLEPN (Mikkelsen, 2014), MTH3 (Tawfik, 2015), PABP_YEAST (Fields, 2013), POLG (Sun, 2014), RL401 (Bolon, 2013), RL401 (Bolon, 2014), UBE4B (Klevit, 2013), YAP1_HUMAN (Fields, 2012))

**B**  DeepSequence   TDVAE

**C**  DeepSequence   TDVAE

**D**  DeepSequence   TDVAE

(Figure 1B–D: scatter/contour plots of Fitness (log) versus $S_m = \log(p(x_{mut})/p(x_{wt}))$ for DeepSequence and TDVAE)

**Fig. 1 | Performance on mutation effect prediction. A** For each protein dataset, we report the performance of TDVAE and DEEPSEQUENCE in terms of Spearman's rank correlation computed over 5 independent runs and using the best parameters associated with the best validation loss. We denote as TDVAE `Low mem` the TDVAE model trained using a mini-batch size of 4 sequences. **B** Correlation between evolutionary scores computed by DEEPSEQUENCE and TDVAE for the *BRCA1* dataset, (**C**) the *BLAT* (Tenaillon, 2013) dataset, and (**D**) the *DLG* dataset; the dashed lines represent the wildtype fitness level in the experimental data (y-axis) and as predicted by the models (x-axis).

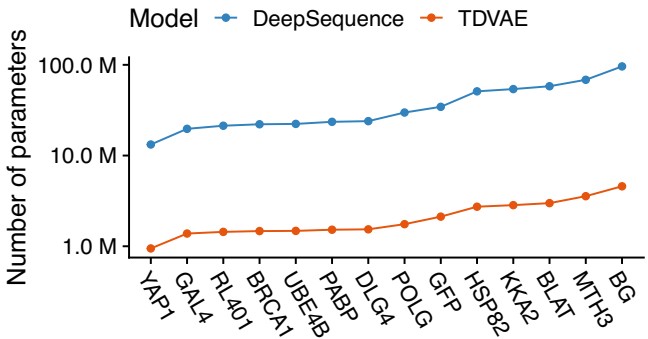

**Fig. 2 | Analysis of TDVAE and DEEPSEQUENCE model complexity.** The x-axis reports the protein analysed in our benchmark dataset, while the y-axis the number of parameters of each model. Here, we found that TDVAE has approximately 94% fewer parameters than DEEPSEQUENCE on average.

Differences in performance between the models could not be apportioned to either the size of the MSA, the number of mutations or the number of homologues used to train the model. However, we observed that both DEEPSEQUENCE and TDVAE had a similar performance trend on the same datasets, suggesting that mutation effect prediction might vary significantly depending on the specific protein and phenotype studied. This becomes apparent when analysing the *BRCA1* dataset (see Fig. 1B), which is the one where both models had their worst performance; specifically, they both report most mutations as detrimental rather than beneficial. Conversely, on the *BLAT* dataset (see Fig. 1C), which is the one with the highest correlation for both models and on the *DLG4* dataset (see Fig. 1D), where we observed the largest TDVAE improvement, both models accurately capture the fitness variability in the dataset.

While we showed that TDVAE achieves comparable state-of-the-art performance, we wanted to test whether these results could be related to our model being more complex than DEEPSEQUENCE. Thus, we computed the number of model parameters for both models on each of the 14 distinct proteins in our benchmark dataset. Here we found TDVAE to be ~ 94% smaller than DEEPSEQUENCE on average, ranging from ~ 945 K parameters for the *YAP1* dataset to ~ 4.6 M parameters for the *BG* dataset (see Fig. 2), suggesting that TDVAE performance cannot be apportioned to model over-parametrization but rather to the combination of efficient encoder and decoder architectures coupled with accurate latent space modelling.

Taken together our results suggest that TDVAE is a robust model for mutation effect prediction, which achieves good performance while being substantially less complex than other approaches.

## TDVAE outperforms Gaussian latent modelling and is robust to parameter setting choices

In our previous experiments we used a TDVAE configuration comparable to the one used by DEEPSEQUENCE, and showed it obtains state-of-the-art performance while requiring the learning of a significantly

smaller number of parameters. We then decided to test how it compares with the same architecture when the latent space is modelled using a canonical Gaussian distribution, a model we dubbed Temporal Gaussian Variational Autoencoder (TGVAE), and whether TDVAE performances were sensitive to parameter choice. To do that, we trained both TDVAE and TGVAE using different configurations; specifically, we varied the number of blocks of stacked dilated convolutions, the intermediate channel size, the latent space dimensionality, and the kernel size. We also tested two scenarios with sub-optimal training procedures, following up on the satisfactory TDVAE LOW MEM performances in the previous experiments by using (i) a mini-batch size of only 8 sequences, and (ii) training for 40, 000 batch updates. Taken together, we constructed 9 different configurations.

We conducted the hyperparameters analysis by training each model configuration on 4 proteins (8 mutagenesis datasets total), namely *YAP1*, *DLG4*, *BRCA1*, and *BLAT*, which vary with respect to the size of the MSA size and initial model performances, in order to work on a small but diverse subset of proteins. Finally, for each model, we computed the best and average performances over 3 independent runs (see Supplementary Data 2).

Experimental results showed that TDVAE is more robust than TGVAE to parameters settings, i.e., performs better on average, for 6 out of the 8 datasets, ranging from 3.10% for YAP1 to 0.02% for the BLAT (Ranganathan, 2015) dataset. Importantly, average performance improvement for TDVAE can be as high as 8.04% for the BRCA-e3 dataset, while performing worse by less than 0.5% on the other 3 datasets on average. A Similar trend was observed when considering the best absolute performance of each model, where TDVAE outperforms TGVAE in 6 out of 8 datasets considered regardless of the configurations used, with an average improvement of $\approx 2\%$, ranging from 0.21% for the DLG4 dataset to 6.42% for the BRCA1-e3 dataset.

Taken together, our analysis shows that TDVAE performs robustly with respect to parameters settings, and that, in general, the use of the Dirichlet distribution is associated with better predictive power, albeit the extent of improvement depends on the specific protein studied.

## TDVAE generates variants of the human α-galactosidase enzyme with wildtype properties

We have already shown that our model can learn the fitness landscape of a protein in an unsupervised fashion. Thus, we hypothesised that TDVAE should be able to generate new variants that are similar at sequence and structural levels with respect to a target wild-type protein. However, evaluation of proteins designed by generative models is usually difficult in silico because experimental mutagenesis datasets usually encompass single-locus mutations, whereas generative models can design very diverse proteins with a potentially high number of mutations. Therefore, as a testbed, we looked for protein coding genes associated with Mendelian diseases and with a well-characterised mutational landscape, such that we can have a more unbiased approach to evaluate in-silico whether the designed variants could be functional.

Here, we focused on the human α-galactosidase (AGAL) lysosomal enzyme (Uniprot id: P06280), a 429 amino acid long protein responsible for hydrolysing the terminal α-galactosyl moieties from glycolipids and glycoproteins[17]. Inherited loss of function mutations in the *GLA* gene, which encodes this enzyme, leads to the accumulation of partially metabolised glycosphingolipids, particularly globotriaosylceramide (Gb3) and globotriaosylsphingosine (lyso-Gb3) in multiple cells. Progressive accumulation of glycosphingolipids, a condition known as Fabry disease, ultimately leads to organ damage, particularly heart and kidney, and premature death[15]. Enzyme replacement therapies (ERTs), which consist of the infusion of a recombinant version of the AGAL enzyme, are the current standard of care. However, current ERTs have poor catalytic activity, are unstable in blood, and often are immunogenic[18,19]. Generating large variant

libraries of AGAL enzymes to screen for desirable therapeutic properties represents an attractive approach to address these issues. However, the α-galactosidase (AGAL) mutational landscape, one the most well-characterised among all lysosomal storage diseases, consists of 216 known validated single point mutations covering more than 50% of the protein sequence, suggesting that designing new recombinant AGAL enzymes is a challenging task.

**Library design.** To design new AGAL variants with TDVAE, we obtained homologue sequences from the UNICLUST30 database using HHBLITS (see "Methods"), which gave us an initial dataset of 1, 746 sequences. We then further processed this dataset to remove sequences with non-canonical amino acids (35 in total), and then split the remaining sequences into training and validation datasets using a 90/10 ratio. We then used TDVAE in a configuration similar to the one used for mutation effect prediction, albeit we introduced a 32-dimensional embedding layer to better deal with variable length sequences. We then used a 32-dimensional Dirichlet latent space, in order to work with a compact latent space, whereas we used a 128-dimensional channel size for the causal convolutional layers and a 20% dropout to prevent over-fitting. Finally, we trained the model for 5, 000 epochs using mini-batches of 256 sequences.

We first tested whether the AGAL sequence landscape could be effectively mapped to a Dirichlet latent space. To do that, we analysed the empirical standard deviation of each latent space component, defined as $\text{diag}(\text{cov}_x[\mathbb{E}_{q_\phi(z|x)}[z]])$[20]. We considered a component as being active if its standard deviation was greater than 0.03, which represents the expected standard deviation if a model assigns random values to a latent component. Here we found TDVAE effectively uses all the 32 components, albeit with a different relative importance, suggesting that different embeddings are used to encode different sequences (see Fig. 3A). We further validated this finding by performing Principal Component Analysis (PCA) of the 32-dimensional expected value parameter of the variational posterior distribution, $q_\phi(z|x)$, for the sequences in the training set. PCA revealed distinct clusters associated with the 3 largest classes found in the training set (mammals, birds and fish), with the emergence of a core common to all embeddings, which is consistent with sequences being all members of the α-galactosidase family (see Fig. 3B).

**Sequence analysis.** After training the model, we generated variants of the wildtype human AGAL enzyme, by first passing the wildtype sequence in input to our encoder to obtain the parameters of the associated region of the Dirichlet latent space, and then sampled 20,000 independent latent vectors to be decoded into sequences by picking the most likely amino acid at each position. Generated sequences were further processed using BLASTP by filtering out those with E-value > 0.001 and query coverage below 75%. All 20,000 samples passed these filters and were then considered for downstream analysis.

Variants have an average of $\approx 48$ mutations (see Fig. 4A), albeit not localised at random but clustered in specific regions of the enzyme (see Fig. 4B). The most variable region is located at the N-terminus (1−32 residues), where the signalling peptide is encoded: this is expected as the signalling peptides change significantly across species[21]. Importantly, the binding site region, spanning residue 203−207, is highly conserved, which confirms that the model is not introducing any obvious inactivating mutations. Using the BLOSUM62 substitution matrix, we analysed the mutations introduced in at least 75% of the generated sequences: here, we found that TDVAE always introduces conservative mutations except in three locus, with the most non-conservative mutation being D233Y (see Fig. 4C), suggesting that our model generates diversity by introducing putative non-detrimental changes.

We then characterised the biochemical properties of our variants and compared them with those of the wildtype enzyme (see Fig. 5A).

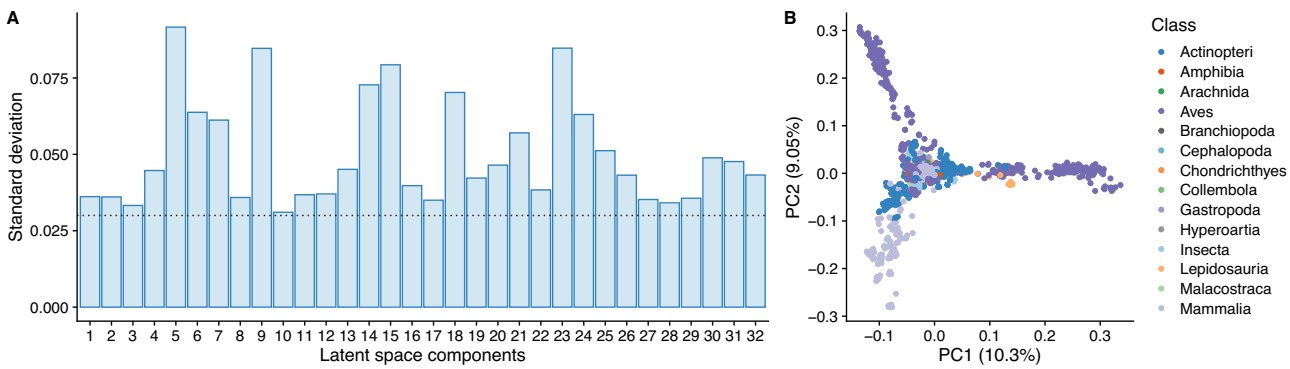

**Fig. 3 | TDVAE learns the landscape of α-galactosidase (AGAL) homologues.**
**A** Activation plot for the 32-dimensional latent space learned by TDVAE. The x-axis represents each latent component, and the *y*-axis is the standard deviation of the corresponding value learned during training; the dashed line represents the

expected standard deviation if the model assigns random values to each component. **B** Principal component analysis of the $\mathbb{E}_{q_\phi(z|x)}[z]$ embeddings generated by TDVAE for the sequences in the training set and labelled according to the corresponding lineage class.

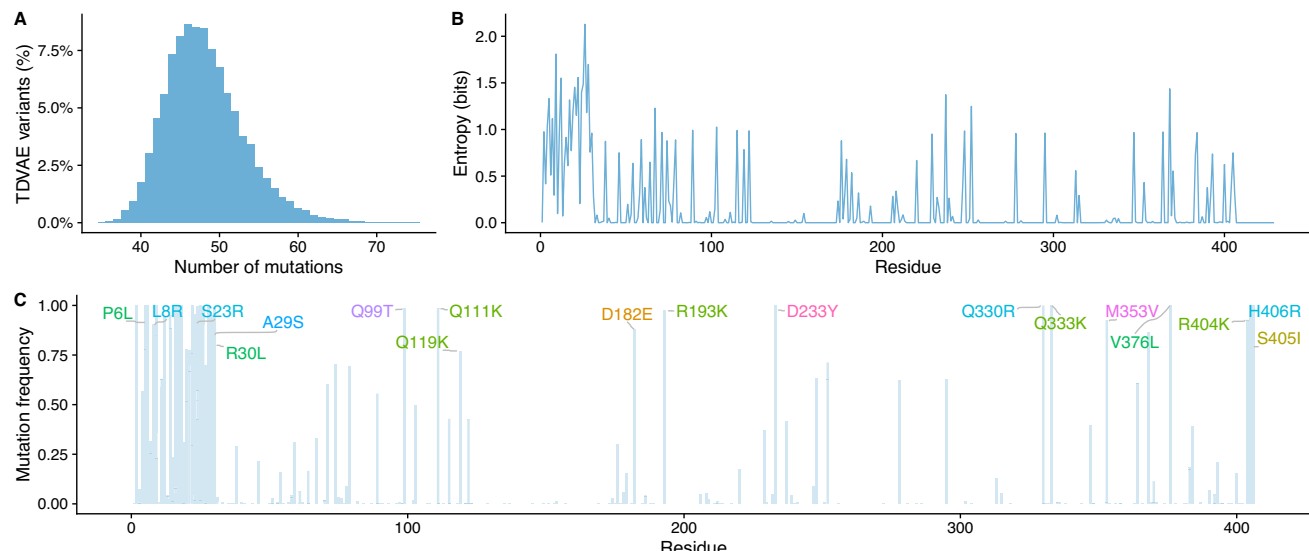

**Fig. 4 | TDVAE generates a diverse library of AGAL variants. A** Distribution of the number of mutations per variant in the library of 20,000 sequences generated by TDVAE. **B** Residue-level entropy of TDVAE variants. **C** Most frequent mutations in TDVAE variants.

Here, we found our variants to have similar molecular weight, flexibility and isoelectric point compared to the wildtype, albeit they are predicted to be more stable, suggesting that our library of variants retains the biochemical features of the wildtype.

We then looked at whether any of the introduced mutations are associated with a significant reduction of enzymatic activity and disease phenotypes. The generated library did not contain pathogenic mutations, albeit 2 mutations were found in more than 5% of the sequences (see Fig. 5B), respectively Q330R and D313N; both mutations were found in Fabry patients[22], but the most frequent one, Q330R, is not associated with a physiologically relevant accumulation of Gb3, whereas the second one has an unclear phenotype; interestingly, both mutations are reported as 'benign' by POLYPHEN2, a standard tool for assessing the impact of protein mutations.

Taken together, our sequence analysis shows that TDVAE can generate a diverse library of AGAL enzymes while retaining the functional features of the wild-type sequence.

**Structural analysis.** We also studied our variant library at the structural level to identify potential functional changes and differences in stability compared to the wild type. To do that, we ranked sequences

by their associated ELBO and selected the top 20 sequences from 5 equally spaced deciles as a way to select diverse variants across the library. We predicted the structure of the 100 selected sequences (see "Methods") and used them for downstream comparative and motion analysis using the human wildtype AGAL structure (PDB id: 1R46) as the reference structure.

PCA showed that TDVAE variants are similar to the known wildtype structure (see Fig. 6A), which is consistent with the average RMSD being ~ 0.87 Å (see Fig. 6B) and the high sequence homology with the wildtype of all the variants. Structural variability in the variants was limited to coiled regions, with root mean square fluctuation (RMSF) less than 1 Å everywhere except at the N-terminus (see Fig. 6C). Finally, we performed ensemble Normal Mode Analysis (eNMA)[23] to probe large scale motion of our variants. Simulation results showed that the designed variants have a flexibility profile consistent with the wildtype protein (see Fig. 7A), albeit variants show major instability around the asparagine residue in position 335 (see Fig. 7C) compared to wildtype (see Fig. 7B), whereas they proved to be more stable around the glutamic acid at position 58.

Taken together, our analyses confirm that TDVAE can generate a diverse library of putatively active AGAL variants while retaining the structural and functional properties of the wild-type enzyme.

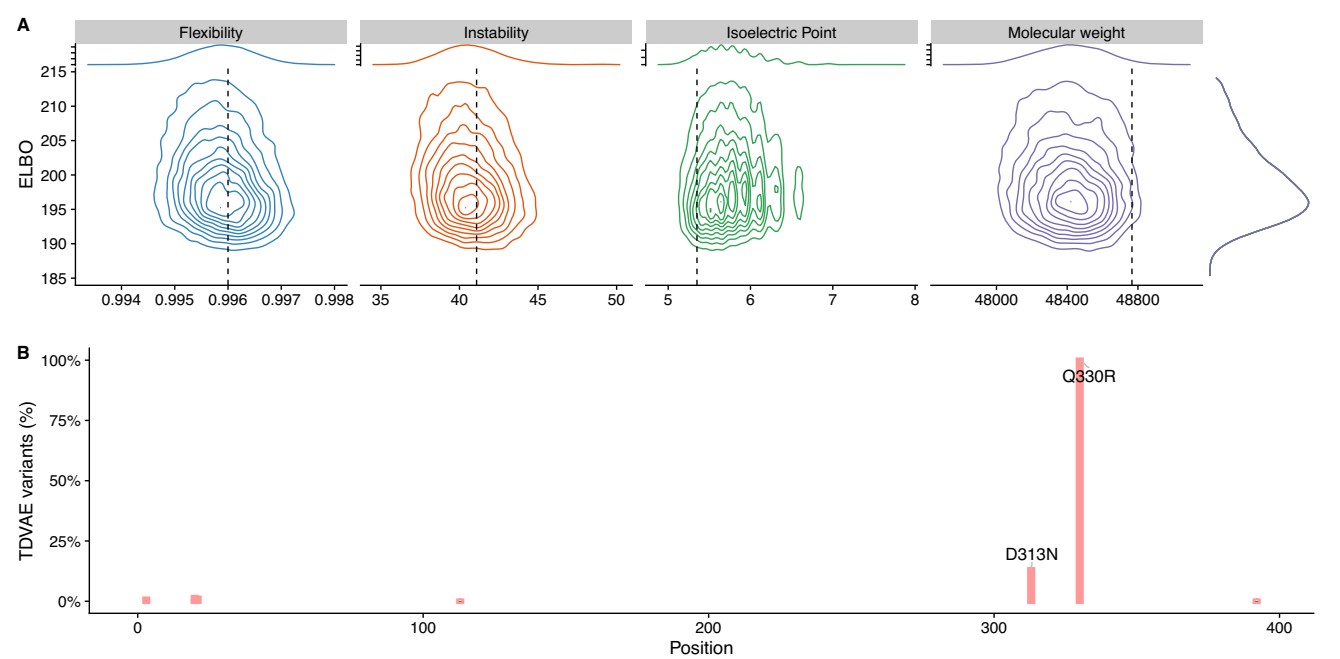

**Fig. 5 | AGAL variants have wild-type-like biochemical properties. A** Analysis of biochemical properties for the TDVAE variant as a function of the associated ELBO. **B** Frequency of mutations in the TDVAE variants associated with Fabry disease or changes in enzymatic activity.

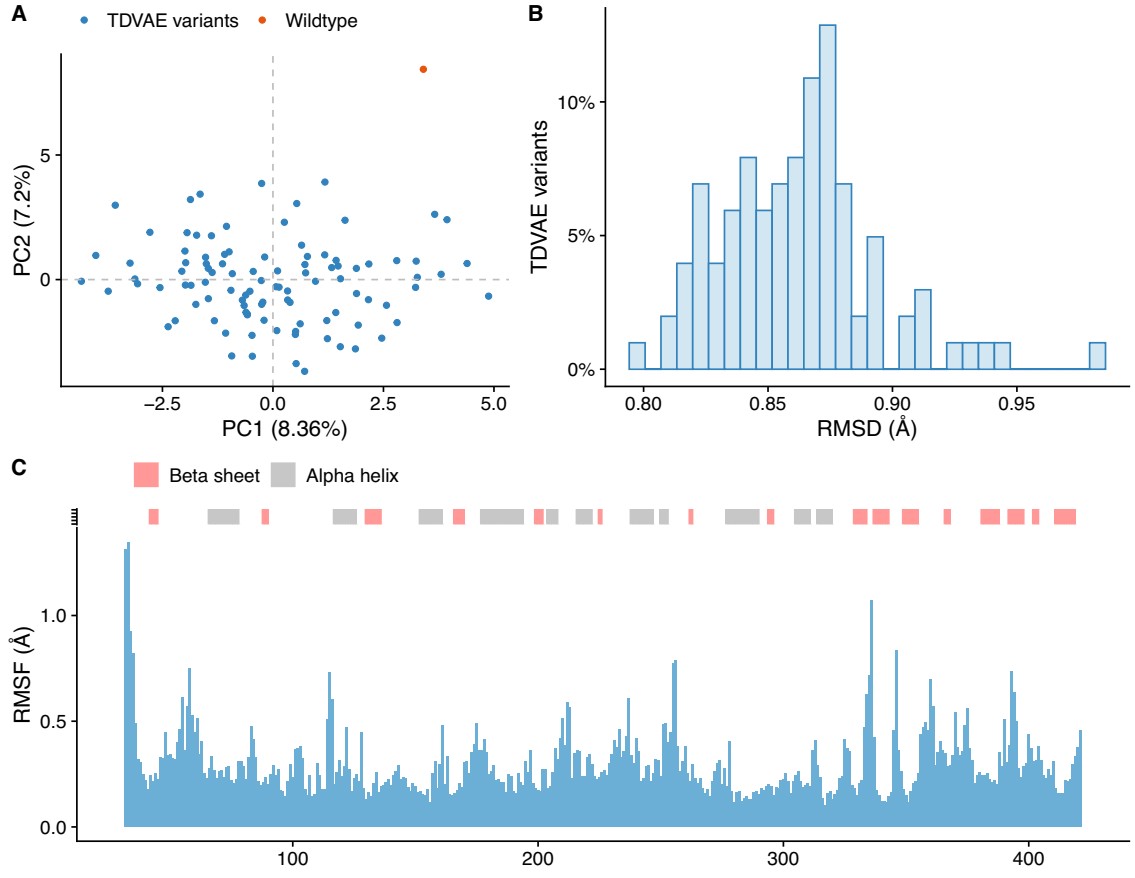

**Fig. 6 | AGAL variants have wildtype-like structures. A** Principal Component Analysis (PCA) analysis of wildtype and variant AGAL structures. **B** Distribution of root mean squared deviations (RMSD) of the variants from the wildtype structure (PDB id: 1R46). **C** Structural variance analysis of TDVAE variants; the *y*-axis reports the average root mean square fluctuation (RMSF) per residue.

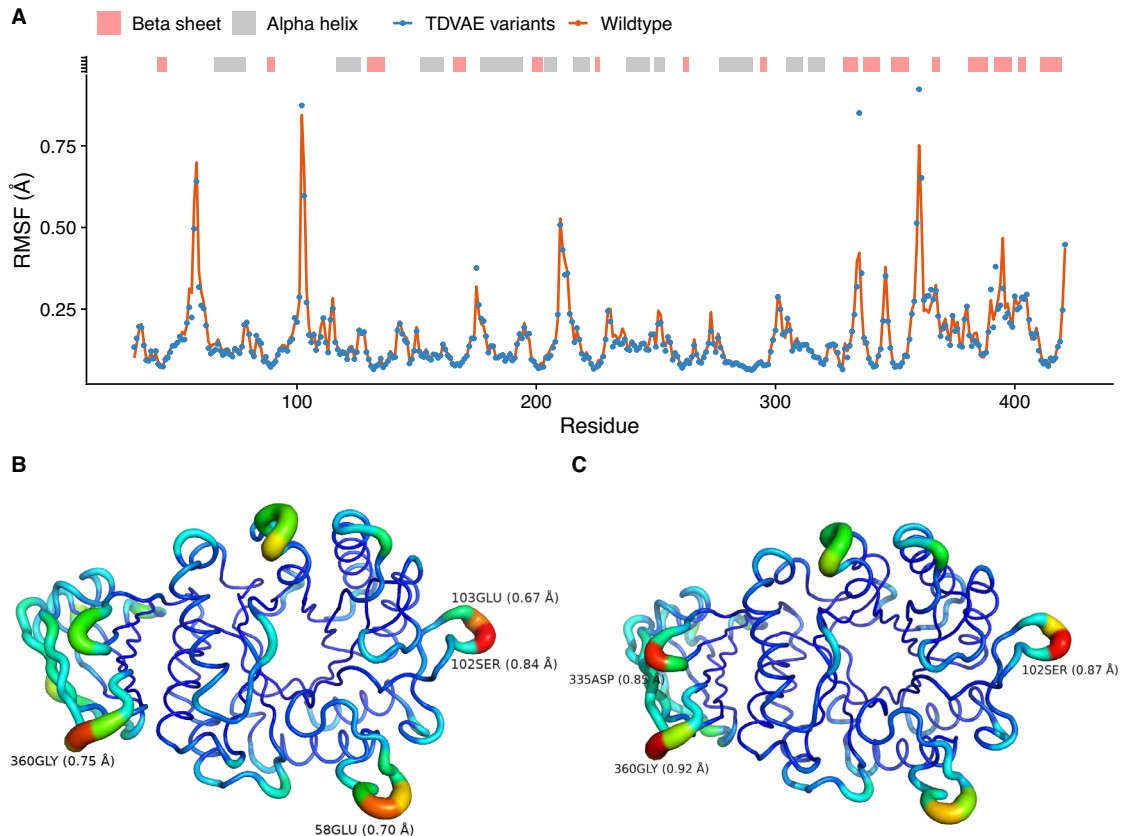

**Fig. 7 | AGAL variants preserve the structural flexibility of the wild-type enzyme. A** ensemble Normal Mode Analysis (eNMA) of the wildtype and TDVAE variants. The RMSF of the variants is averaged per residue across the library.

**B** RMSF from eNMA analysis for the wildtype structure. **C** Average RMSF from eNMA analysis for the variants library projected on the wildtype structure.

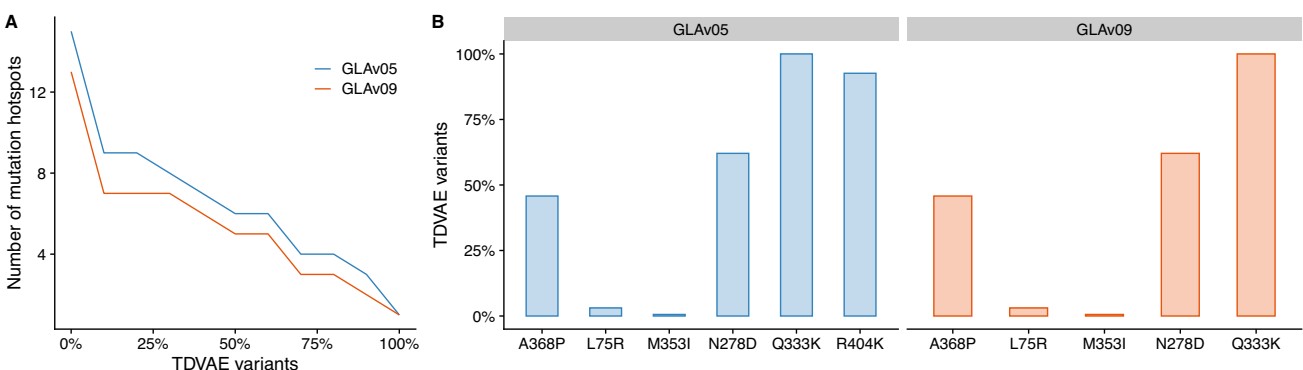

**Fig. 8 | TDVAE identifies beneficial mutational hotspots. A** Percentage of variants harbouring mutations at the mutational hotspots identified in GLAv05 and GLAv09[24]. **B** Percentage of TDVAE variants carrying GLAv05 and GLAv09 beneficial mutations.

**Comparison with directed evolution.** The ability of TDVAE of generating enzymes with wildtype features represents an attractive approach to increase the success rate of variant library screening workflows, where randomised approaches, such as directed evolution, are the industry standard.

To quantify the potential benefit of onboarding our approach, we compared our results with a recent directed evolution study which identified, out of 12, 000 screened enzymes, 2 variants, GLAv05 and GLAv09, with higher catalytic activity and stability and lower immunogenicity[24]. These two enzymes identify a total of 19 loci harbouring beneficial mutations, which we denote as mutational hotspots. We used this information to assess whether mutations at

hotspots were enriched in our library of variants, under the assumption that the higher the number of variants harbouring mutations at the hotspots and matching the same beneficial mutations, the better the library is. Here, we found 2 loci mutated in at least 90% of the sequences and 7 in at least 50% of them (see Fig. 8A), suggesting that TDVAE identified most of the mutational hotspots in an unsupervised fashion from sequence information alone. We then looked at the frequency of the beneficial mutations in GLAv05 and GLAv09, and found 2 beneficial mutations, Q333K and R404K, in at least 90% of the variants, and 4 in ~50% of them (see Fig. 7D), suggesting that our model has the potential to identify not only the hotspots but also the specific beneficial mutations.

Finally, we used this data to further compare our library with another library generated by TGVAE of the same size using the same training strategy. Interestingly, we found that TGVAE performs worse than TDVAE, with less than 25% of its variants having mutations at the hotspots and less than 1% of the variants having exact GLAv05 and GLAv09 mutations, a significantly worse library compared to the one designed by TDVAE (see Supplementary Fig. 1).

In conclusion, our results provide strong evidence of the benefits of using TDVAE as a tool to optimise the design of variants library.

## Discussion

Learning how an amino acid sequence can be engineered to obtain a desired biochemical function remains an open challenge in biology, biotechnology and medicine. Recent developments in deep learning are now allowing us to learn the biochemical rules to engineer new functional proteins.

Here, we introduced a deep learning model called Temporal Dirichlet Variational Autoencoder (TDVAE), to effectively and efficiently predict the effect of amino acid mutations and design new variants of known proteins. Since TDVAE is an unsupervised deep autoregressive model, these tasks are achieved without experimental data but by extracting higher-order information from a functionally related set of sequences. Therefore, TDVAE represents an attractive approach to effectively design large protein libraries to be screened to identify candidates with desired properties. While the use of deep autoregressive models is not new, we introduced a new architecture integrating a Dirichlet latent space with Temporal Convolutional Networks (TCNs) as a more accurate and efficient approach to estimating amino acid distributions compared to standard Gaussian latent space modelling.

To prove that, we first used TDVAE as a mutation effect prediction tool, and found that our model achieves excellent performances while being significantly less taxing to train and use than state-of-the-art models. We then conducted a hyperparameter analysis to further validate the use of the Dirichlet distribution for latent modelling and showed that it leads to better results when compared to the Gaussian distribution. Taken together, we hypothesise that such performance improvement could be apportioned to Dirichlet distribution being more suitable for capturing the multi-modality of the protein fitness landscape.

Then, we showed that TDVAE is also able to design variants of a complex protein, like the human $\alpha$-galactosidase (AGAL), while retaining wildtype features at sequence, structural and functional levels, which are essential to use this enzyme as a potential enzyme replacement therapy for Fabry disease. Remarkably, we showed that our model identified mutational hotspots and beneficial mutations exclusively in-silico, whereas it required screening more than 12,000 variants using directed evolution[24]. Therefore, our analysis suggests that TDVAE could be a powerful tool to maximise the success rate of protein engineering workflows by designing libraries of functional proteins for massively parallel screening.

Despite our model holds the promise of being an effective protein engineering and analysis tool, we are also aware of its limitations. First, despite generating wild-type proteins, identifying those with desired characteristics requires downstream post-processing and experimental validation; extending TDVAE with a conditional generative learning process can address this, albeit at the cost of increased model complexity and computational burden for training and inference. While sequence information has been shown to be sufficient to tackle many protein engineering-related tasks, introducing structural information will likely be beneficial, especially in designing enzymes and protein-ligand complexes, where physical constraints are key to obtaining the desired function[25,26].

Taken together, we anticipate that the introduction of a robust, unsupervised, sequenced-based model like TDVAE will allow us to take advantage of the increasing number of available protein sequences and structures across the kingdom of life, to deepen our understanding of the biochemical rules to engineer functional designer proteins.

## Methods

### A deep generative learning framework for protein engineering

Protein engineering requires learning how to sample the protein design space to identify amino acid sequences associated with a desired catalytic function. Here, we hypothesise that the design space has a statistical structure whose functional form and corresponding parameters are unknown but can be learned from known sequences readily available in protein databases.

We hereby assume that the probability of observing a protein sequence $x$ depends on a latent random variable $z$, such that:

$$p_{\theta}(x, z) = p_{\theta}(x|z)p_{\theta}(z) \tag{1}$$

where $p_{\theta}(x|z)$ and $p_{\theta}(z)$ are parametric distributions. Thus, a protein sequence can be considered the result of a generative process, which involves sampling a random variable $\hat{z}$ from $p_{\theta}(z)$, which in turn is used to build a sequence $\hat{x}$ by sampling from the conditional probability distribution $p_{\theta}(x|\hat{z})$; in our case, $\hat{z}$ can be thought as a random variable encoding properties specific to the proteins in the training dataset, such as function or amino acid composition.

However, learning the parameters $\theta$ of this class of models is usually intractable, since we cannot evaluate or differentiate the marginal likelihood $\int p_{\theta}(x|z)p_{\theta}(z)dz$ and the posterior probability $p_{\theta}(z|x) := \frac{p_{\theta}(x|z)p_{\theta}(z)}{p_{\theta}(x)}$. Here we addressed these issues by using a Variational Autoencoder (VAE) architecture, where Neural Networks (NNs) are used to approximate $p_{\theta}(x|z)$ and $p_{\theta}(z|x)$, and Stochastic Variational Inference (SVI) to learn model parameters[5]. Specifically, in a VAE framework, $p_{\theta}(z|x)$ is approximated by a parametric recognition model, $q_{\phi}(z|x)$, which acts as a probabilistic encoder taking in input a sequence $x$ and returning a distribution over the possible value of $z$. Thus, $p_{\theta}(x|z)$ acts as a probabilistic parametric decoder, which takes in input a sample $z$ and returns a distribution over the possible values of $x$.

Here, we argued that the ability of VAEs to effectively sample the protein design space and generate new functional variants depends on the parametric family used to model the latent space and the use of Neural Networks (NNs) for computing $p_{\theta}(x|z)$ and $q_{\phi}(z|x)$.

VAEs have traditionally assumed the prior distribution of the latent space to be continuous, ultimately leading to the ubiquitous use of a Gaussian prior distribution. However, protein sequences inherently follow a multivariate multinomial distribution, which represents the probability of observing a given amino acid at any given position. Importantly, the conjugate prior of the multinomial distribution is the Dirichlet distribution, thus modelling a discrete latent space represents a necessary step to build effective VAEs for sequence modelling. Moreover, from an engineering perspective, the design space is expected to be highly multimodal, as a result of the biophysical forces controlling protein folding; this multimodality cannot be modelled by a multivariate Gaussian distribution[27], thus we hypothesised that modelling the latent space using a Dirichlet prior could be beneficial in generating functional proteins.

Computing $p_{\theta}(x|z)$ and $q_{\phi}(z|x)$ over sequences are usually intractable, thus a plethora of NNs have been proposed as robust approximations, including Recurrent Neural Network (RNN)[28], Long Short Term Memory (LSTM)[29] and Gated Recurrent Unit (GRU)[30]; however, as the length of the sequences increases, their ability to learn long-range relationships between amino acids decreases[31], a drawback that makes them unsuitable for handling long amino acid sequences. Moreover, these architectures are computationally expensive to

train[32], as they cannot be readily parallelised, and thus unsuitable to scale over large sequence datasets.

Therefore, we used an alternative architecture for both the encoder and decoder, called Temporal Convolutional Network (TCN), which overcomes these limitations and can be efficiently trained[14]. TCNs take sequences in input and return new ones of the same length.

By utilising a standard 1-dimensional convolutional layer with zero padding on the left, we can ensure that the output size in the subsequent layers matches the input size. To condition the probability of a residue on the previously observed ones, TCNs use causal convolution. This technique computes the residue at position $t$ by applying the convolution only with the elements at position $t$ down to 0 in the previous layer. However, achieving a full sequence coverage usually requires stacking a number of convolutional layers proportional to the sequence length, thus making standard TCNs inefficient to train for long sequences. The problem has been mitigated by using dilated convolution. For a sequence $\boldsymbol{x}$ of length $n$ and a kernel $\boldsymbol{f}$ of size $k$, the dilated convolution $\boldsymbol{F}$ is calculated as:

$$\boldsymbol{F}(s) = (\boldsymbol{x}^*_d\boldsymbol{f})(s) = \sum_{i=0}^{k-1} \boldsymbol{f}(i) \cdot \boldsymbol{x}_{s-di} \qquad (2)$$

Here, $d$ represents the dilation factor, which determines the spacing between input elements that are used to compute one element of the output. By stacking multiple temporal convolution layers with exponentially increased dilation factors, full sequence coverage can be achieved with the number of layers logarithmic in sequence length. Specifically, to reduce the number of hyperparameters to optimise, we determined the total number of stacked layers as follows:

$$n = \left\lceil \log_2 \left[ \frac{(L-1)}{2(k-1)} + 1 \right] \right\rceil \qquad (3)$$

where $L$ is the length of the longest sequence.

**Variational inference of model parameters.** In a Variational Autoencoder framework, NNs are used to compute the variational parameters $\boldsymbol{\phi}$ for a fixed family of probability distributions $q_{\boldsymbol{\phi}}(\boldsymbol{z}|\boldsymbol{x})$ and model parameters $\boldsymbol{\theta}$ for conditional likelihood $p_{\boldsymbol{\theta}}(\boldsymbol{x}|\boldsymbol{z})$. Here, we use SVI to find an approximate solution to the problem of maximising the marginal likelihood $\int p_{\boldsymbol{\theta}}(\boldsymbol{x}|\boldsymbol{z})p_{\boldsymbol{\theta}}(\boldsymbol{z})d\boldsymbol{z}$ by maximising the Evidence Lower Bound (ELBO) w.r.t both model parameters $\boldsymbol{\theta}$ and variational parameters $\boldsymbol{\phi}$ as follows:

$$\mathcal{L}(\boldsymbol{\phi}, \boldsymbol{\theta}) = \mathbb{E}_{q_{\boldsymbol{\phi}}(\boldsymbol{z}|\boldsymbol{x})}[\log p_{\boldsymbol{\theta}}(\boldsymbol{x}|\boldsymbol{z})] - KL(q_{\boldsymbol{\phi}}(\boldsymbol{z}|\boldsymbol{x})||p_{\boldsymbol{\theta}}(\boldsymbol{z})) \rightarrow \max_{\boldsymbol{\theta}, \boldsymbol{\phi}} \qquad (4)$$

where KL is the Kullback-Leibler divergence, and the expected conditional likelihood $\mathbb{E}_{q_{\boldsymbol{\phi}}(\boldsymbol{z}|\boldsymbol{x})}[\log p_{\boldsymbol{\theta}}(\boldsymbol{x}|\boldsymbol{z})]$ is optimised with respect to maximum log-likelihood, as the probability distribution $p_{\boldsymbol{\theta}}(\boldsymbol{x}|\boldsymbol{z})$ over the amino acid space is categorical.

Depending on the choice of parametric families for $q_{\boldsymbol{\phi}}(\boldsymbol{z}|\boldsymbol{x})$ and $p_{\boldsymbol{\theta}}(\boldsymbol{x}|\boldsymbol{z})$, computing the expected conditional likelihood can be challenging, is often intractable and requires numerical approximations, whereas KL can be computed analytically. Ultimately, we need to compute a gradient of the expected conditional likelihood w.r.t parameters $\boldsymbol{\phi}$: $\nabla_{\boldsymbol{\phi}} \mathbb{E}_{q_{\boldsymbol{\phi}}(\boldsymbol{z}|\boldsymbol{x})}[\log p_{\boldsymbol{\theta}}(\boldsymbol{x}|\boldsymbol{z})]$. However, in this case, the gradient computation cannot be moved under the expectation operator, since the expectation is done w.r.t $q_{\boldsymbol{\phi}}(\boldsymbol{z}|\boldsymbol{x})$. Nonetheless, for many parametric distributions, a number of low variance gradient estimators have been proposed, such as those based on pathwise derivatives, alternatively known as reparametrization trick[5,33], but they cannot be applied to the Dirichlet distribution unless Gaussian-based approximations are used at the cost of losing the characteristic properties of the Dirichlet distribution[27]. The general idea of the generalised reparametrization

trick that applies to the majority of continuous distributions is based on implicit differentiation that results in $\nabla_{\boldsymbol{\phi}}\boldsymbol{z}$ term that can be computed using only Probability Density Function (PDF) $q_{\boldsymbol{\phi}}(\boldsymbol{z})$ and derivatives of Cumulative Density Function (CDF) $\nabla_{\boldsymbol{\phi}}F(\boldsymbol{z}|\boldsymbol{\phi})$ or its numerical approximation as follows:

$$\nabla_{\boldsymbol{\phi}}\mathbb{E}_{q_{\boldsymbol{\phi}}(\boldsymbol{z})}[f(\boldsymbol{z})] = \mathbb{E}_{q_{\boldsymbol{\phi}}(\boldsymbol{z})}[\nabla_{\boldsymbol{z}}f(\boldsymbol{z})\nabla_{\boldsymbol{\phi}}\boldsymbol{z}]$$
$$\nabla_{\boldsymbol{\phi}}\boldsymbol{z} = -\frac{\nabla_{\boldsymbol{\phi}}F(\boldsymbol{z}|\boldsymbol{\phi})}{q_{\boldsymbol{\phi}}(\boldsymbol{z})} \qquad (5)$$

The exact derivation of $\nabla_{\boldsymbol{\phi}}\boldsymbol{z}$ for Dirichlet distribution is provided in refs. 34,35, with the latter being an official implementation of the PyTorch library.

**Data collection.** Homologous sequences for the human $\alpha$-galactosidase (AGAL) enzyme where obtained by searching the UNICLUST30 database using HHBLITS[36] with parameters `-Z 10000 -B 10000 -e 0.001 -all` and filtered to retain only sequences with 80% query coverage and 50% query identity (`-id 100 -cov 80 -qid 50`). The final dataset consisted of 1746 sequences, from which we removed gaps and insertions.

**AGAL library analysis.** Biochemical properties of the variants in the AGAL library were all computed using the `ProtParam` module of the BIOPYTHON package[37]. Structures for selected variants of the library where predicted using ESMFOLD[38] and successively relaxed by energy minimisation as implemented in the OPENMM package[39] using the Amber14 force field and adding an harmonic potential energy term to restrain C$\alpha$ atoms position. The downstream normal mode analysis was performed using the BIO3D package[40].

**Reporting summary**

Further information on research design is available in the Nature Portfolio Reporting Summary linked to this article.

## Data availability

The data supporting the findings of this study are available on Zenodo at: https://doi.org/10.5281/zenodo.13269310.

## Code availability

The software is available under an Academic-only license at the following https://licensing.edinburgh-innovations.ed.ac.uk/product/proton.

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

## Acknowledgements

This work was supported by the UKRI EPSRC Fellowship (EP/V033794/1) and EPSRC funding (EP/Y01913X/1) to G.S and the UKRI Centre for Doctoral Training in Biomedical AI CDT (EP/S02431X/1) for E.L. Computational experiments were performed using resources provided by the Cambridge Service for Data-Driven Discovery (CSD3) operated by the University of Cambridge Research Computing Service (www.csd3.cam.ac.uk), provided by Dell EMC and Intel using Tier-2 funding from the Engineering and Physical Sciences Research Council (capital grant EP/P020259/1), and DiRAC funding from the Science and Technology Facilities Council (www.dirac.ac.uk).

## Author contributions

G.S. conceived the study. G.S. and E.L. formulated and developed the model. E.L. implemented and tested the model and performed sequence analysis under G.S. supervision. G.S. and E.L. wrote the manuscript.

## Competing interests

G.S. is the founder and CEO of ZYTHERA. The other authors declare no competing interests.
