## [Peer Review File · Nature Communications]

Dirichlet latent modelling enables effective learning and sampling of the functional protein design spaceReviewer #1 (Remarks to the Author):

In this study, the authors propose a new autoregressive model called Temporal Dirichlet Variational Autoencoder (TDVAE) that utilizes the mathematical properties of the Dirichlet distribution to capture high-order information from functionally related sequences, even those that are remotely similar. The authors assert that their method significantly outperforms other existing methods in predicting the effects of mutations and has the potential to design variants of human alpha-galactosidase enzymes for treating Fabry disease. However, the results obtained in this study do not provide enough evidence to support their claim.

Comments:

(1) Accuracy is the primary consideration when evaluating methods. In their comparison with DeepSequence on 19 datasets (Figure 1A), the authors claim superior performance of their method on most datasets compared to DeepSequence. However, a closer examination reveals that TDVAE outperforms DeepSequence in terms of Spearman correlation coefficient on only approximately 8 out of 19 datasets, without conducting significant comparisons to determine the statistical significance of this improvement. Moreover, in the literature from which the 19 datasets used by the authors were derived [16], a simple augmented approach has already been proposed that significantly enhances the predictive performance of several popular methods, including DeepSequence. This approach is referred to as Augmented DeepSequence VAE. However, the authors did not compare their method with this approach, despite using the same 19 datasets constructed by it. Furthermore, the authors claim that the TDVAE model is robust, but this claim is not supported by the provided research results.

(2) The description of the Methods section in the paper is overly brief, focusing solely on the deep generative learning framework, without providing detailed information about the training dataset used for their method. Additionally, although the 19 datasets used by the authors are sourced directly from the literature [16], a brief introduction to these datasets should have been included in the paper, along with an explanation of how their model was applied to these datasets.

(3) Regarding the example of the human α -galactosidase (AGAL) enzyme provided by the authors, the explanation appears oversimplified and raises several concerns. The definition of mutational hotspots is too simplistic, as the two variants (GLAv05 and GLAv09) mentioned have higher catalytic activity, stability, and lower immunogenicity, but it does not imply that all mutations in these variants are beneficial. Therefore, further research is required to determine whether all loci harboring mutations in these variants can be classified as mutational hotspots. Furthermore, the statement "Here we found 2 loci mutated in at least 90% of the sequences and 7 in at least 50% of them (see Figure 8A), suggesting that TDVAE identified most of the mutational hotspots in an unsupervised fashion from sequence information alone" is misleading. Although the authors define 19 mutational hotspots, they only mention two loci and claim that most of the mutational hotspots have been identified.

(4) The authors did not provide the source code and the implementation details of their method. It is necessary to provide the source code for reproducibility in methodological research.

Reviewer #2 (Remarks to the Author):

Remarks to Authors:

The authors present an autoregressive model that uses a new distribution (the Dirichlet distribution) to learn high-order information instead of the commonly used standard multivariate Gaussian distribution. The results currently presented are overstated and the model comparisons and evaluation are very weak and not convincing. The primary appeal for this approach is that it is less complex and less taxing to train. However, the model performance on mutation effect predictions is comparable, if not marginally worse, than current state-of-the-art models (e.g., DeepSequence). Importantly, the code is not publicly available making it impossible to adequately evaluate the method or claims in this manuscript. As it currently stands, the paper is not ready for publication at Nature Communications. I encourage the authors to factor in the following main points of feedback before re-submitting.

****Main points of feedback****

1. Novelty and Significance

The use of the Dirichlet distribution for learning high-order information is an interesting idea. However, the paper's current presentation tends to overstate its significance without providing sufficient evidence to support the claims made. I suggest providing a more balanced discussion of the model's novelty and clearly highlighting its unique advantages over existing approaches.

2. Evaluation against DMS

The evidence presented in the paper appears to be relatively limited, as the authors only report the performance on 19 assays. To strengthen the findings, I recommend including evaluations on larger benchmark datasets, such as those curated by previous works [1,2,3], which offer more comprehensive and diverse test cases for assessment. Incorporating these richer benchmarks will enhance the robustness and persuasiveness of the conclusions drawn from this study.

3. Evaluation against biochemical and structural metrics

The evaluation presented in this section lacks sufficient persuasiveness and appears to be somewhat tautological, as the model's training relies on an alignment that inherently incorporates conserved regions and amino acid exchangeabilities. To strengthen this evaluation, it is crucial to include alternative evaluation methods that can truly assess the model's ability to generalize and provide meaningful insights beyond the training data's inherent characteristics.

3. Unjustified focus on Dirichlet latent modeling

The emphasis on the Dirichlet distribution as the main factor for potential improvement is not adequately supported, as the authors have not demonstrated that the Dirichlet latent space, rather than the TCN architecture, is responsible for any observed enhancements. To address this concern, it is crucial for the authors to compare the performance of a TCN with a typical multivariate Gaussian latent space against their proposed model. This evaluation would help establish the relative contribution of the Dirichlet distribution in driving any potential performance gains. Additionally, it is important to note that the evaluation of the latent components revealed that most exhibit only marginal improvements over random, suggesting a need for further investigation and potential refinement of the model.

****Minor points of feedback****

Line 116 – 119 is confusing, redundant, and not true. By looking at figure 1a, DeepSequence outperforms TDVAE in at least 8 datasets, and similarly outperformed TDVAE (low mem) in at least 8 datasets.

Figure 1a – In practice, one would compare the DMS against the mean over the 5 replicates. If you did that where would the correlation fall?

Figure 1b-d The protein names should be included in the figure.

Why does the low mem model outcompete the full model?

Line 146 - What is the meaning of robust in this context?

Line 165 – 169 – confusing

References

- [1] Riesselman, Adam J., John B. Ingraham, and Debora S. Marks. 2018. “Deep Generative Models of Genetic Variation Capture the Effects of Mutations.” *Nature Methods* 15 (10): 816–22.
- [2] Notin, Pascal, Mafalda Dias, Jonathan Frazer, Javier Marchena Hurtado, Aidan N. Gomez, Debora Marks, and Yarin Gal. 2022. “Tranception: Protein Fitness Prediction with Autoregressive Transformers and Inference-Time Retrieval.” In *Proceedings of the 39th International Conference on Machine Learning*, 16990–17. PMLR.
- [3] Rubin, Alan F., Joseph K. Min, Nathan J. Rollins, Estelle Y. Da, Daniel Esposito, Matthew Harrington, Jeremy Stone, et al. 2021. “MaveDB v2: A Curated Community Database with over Three Million Variant Effects from Multiplexed Functional Assays.” <https://doi.org/10.1101/2021.11.29.470445>.

Response to reviewers

Reviewer #1

The authors thank the reviewer for the useful comments. We have now revised our paper to have a more balanced discussion of the performances of our method and clarified its scope, advantages and disadvantages. We have further clarified how the generative ability has been assessed and improved the results interpretation.

We provided detailed answers to each point raised below:

(1) Accuracy is the primary consideration when evaluating methods. In their comparison with DeepSequence on 19 datasets (Figure 1A), the authors claim superior performance of their method on most datasets compared to DeepSequence. However, a closer examination reveals that TDVAE outperforms DeepSequence in terms of Spearman correlation coefficient on only approximately 8 out of 19 datasets, without conducting significant comparisons to determine the statistical significance of this improvement. Moreover, in the literature from which the 19 datasets used by the authors were derived [16], a simple augmented approach has already been proposed that significantly enhances the predictive performance of several popular methods, including DeepSequence. This approach is referred to as Augmented DeepSequence VAE. However, the authors did not compare their method with this approach, despite using the same 19 datasets constructed by it. Furthermore, the authors claim that the TDVAE model is robust, but this claim is not supported by the provided research results.

We have revised the section and clarified the benchmarking protocol; importantly, we have extended our study by performing a model parameters analysis that accounts for different latent distributions, i.e. Dirichlet vs Gaussian, and different parameters settings. The results now show that Dirichlet is robust to different parameters settings and can provide substantial improvements albeit, as expected, this varies depending on the protein. We have released a supplementary file with all the simulations results.

We also do acknowledge that data augmentation can be potentially useful to improve model performances, but it is not relevant to our work that is methodological and focused on assessing the contribution of latent distributions and model architecture.

(2) The description of the Methods section in the paper is overly brief, focusing solely on the deep generative learning framework, without providing detailed information about the training dataset used for their method. Additionally, although the 19 datasets used by the authors are sourced directly from the literature [16], a brief introduction to these datasets should have been included in the paper, along with an explanation of how their model was applied to these datasets.

We would like to point out that an introduction to the dataset was already present on the initial version of the manuscript, which explained how it was used and processed (see Section "TDVAE performance on predicting protein mutation effects").

(3) Regarding the example of the human α -galactosidase (AGAL) enzyme provided by the authors, the explanation appears oversimplified and raises several concerns. The definition of mutational hotspots is too simplistic, as the two variants (GLAv05 and GLAv09) mentioned have higher catalytic activity, stability, and lower immunogenicity, but it does not imply that all mutations in these variants are beneficial. Therefore, further research is required to determine whether all loci harboring mutations in these variants can be classified as mutational hotspots. Furthermore, the statement "Here we found 2 loci mutated in at least

90% of the sequences and 7 in at least 50% of them (see Figure 8A), suggesting that TDVAE identified most of the mutational hotspots in an unsupervised fashion from sequence information alone" is misleading. Although the authors define 19 mutational hotspots, they only mention two loci and claim that most of the mutational hotspots have been identified.

We would like to address this comment since it might have been causing confusion in reviewer's assessment. The AGAL variants GLAv05/v09 were found experimentally by directed evolution by Hallows et al., 2023; the claim about higher activity, lower immunogenicity and higher stability are based on their experimental findings, not on our computational prediction. The authors further concluded that these mutations were beneficial as they lead to the desired phenotypes, albeit with a varying level of characterisation. In our work, we asked whether our model could blindly identify the loci harbouring these beneficial mutations, and whether it could find any of the beneficial mutations found in GLAv05/09 and at what frequency. This is a critical point since it allows a robust validation of computational predictions with experimental results. We have now further extended this section by showing that on the same input, using a multivariate Gaussian distribution, we obtain significantly worse results, in terms of hotspots recovered and beneficial mutations identified; we believe this result further confirms the potential of our Dirichlet approach.

(4) The authors did not provide the source code and the implementation details of their method. It is necessary to provide the source code for reproducibility in methodological research.

We have now provided source code for review only under the University of Edinburgh licensing scheme.

Reviewer #2 (Remarks to the Author):

The authors thank the reviewer for the useful comments that allowed us to improve our manuscript. We have now revised our paper to have a more balanced analysis of the performances of our method and clarified its scope, advantages and disadvantages. We have further clarified how the generative ability has been assessed and improved the results interpretation.

Remarks to Authors:

We provided detailed answers to each point raised below:

****Main points of feedback****

1. Novelty and Significance

The use of the Dirichlet distribution for learning high-order information is an interesting idea. However, the paper's current presentation tends to overstate its significance without providing sufficient evidence to support the claims made. I suggest providing a more balanced discussion of the model's novelty and clearly highlighting its unique advantages over existing approaches.

We thank the reviewer for the comments, which we have now addressed by: 1) perform a direct comparison of TDVAE and the same architecture using a Gaussian latent distribution, which we dubbed TGVAE. We show that TDVAE is more robust to parameters settings, and that performance improvement can be substantial, albeit it varies depending on the protein. 2) We used TGVAE to redesign the AGAL protein using the same input as TDVAE and found that it could not find neither hotspots nor the beneficial mutations found by directed evolution by Hallows et al. 2023. We think this provides a comprehensive assessment of the importance of proper latent modelling.

Finally, we have also stressed the main advantage of our model from a computational point of view, as it's significantly smaller and faster to train than any other.

2. Evaluation against DMS

The evidence presented in the paper appears to be relatively limited, as the authors only report the performance on 19 assays. To strengthen the findings, I recommend including evaluations on larger benchmark datasets, such as those curated by previous works [1,2,3], which offer more comprehensive and diverse test cases for assessment. Incorporating these richer benchmarks will enhance the robustness and persuasiveness of the conclusions drawn from this study.

We thank the reviewer for the comments. We would like to present our results on the 19 datasets used as they are widely used in literature and strike a balanced trade-off between sequence/assay diversity and time required for carrying out extensive analyses. While there is value in testing on other datasets, large scale benchmarking is outside of the scope of this work, since mutation effect prediction represent only one line of evidence to support the validity of our approach.

3. Evaluation against biochemical and structural metrics

The evaluation presented in this section lacks sufficient persuasiveness and appears to be

somewhat tautological, as the model's training relies on an alignment that inherently incorporates conserved regions and amino acid exchangeabilities. To strengthen this evaluation, it is crucial to include alternative evaluation methods that can truly assess the model's ability to generalize and provide meaningful insights beyond the training data's inherent characteristics.

We thank the reviewer for the comments, and we would like to point out that our design results have been validated not only by looking at biochemical properties of the proteins, but also 1) by checking whether our variants carry known detrimental mutations, which is an assessment metric orthogonal to the input, since this information would not be encoded in the input data, but it comes from genetics studies. 2) we also conducted dynamics analysis, which allowed us to assess whether the dynamic properties of our variants are similar to the one of the wildtype, which is critical for enzyme analysis and again an orthogonal metric for design evaluation.

3. Unjustified focus on Dirichlet latent modeling

The emphasis on the Dirichlet distribution as the main factor for potential improvement is not adequately supported, as the authors have not demonstrated that the Dirichlet latent space, rather than the TCN architecture, is responsible for any observed enhancements. To address this concern, it is crucial for the authors to compare the performance of a TCN with a typical multivariate Gaussian latent space against their proposed model. This evaluation would help establish the relative contribution of the Dirichlet distribution in driving any potential performance gains. Additionally, it is important to note that the evaluation of the latent components revealed that most exhibit only marginal improvements over random, suggesting a need for further investigation and potential refinement of the model.

We thank the reviewers for the feedback and we have now compared and contrasted TDVAE performances against the same architecture but using a latent multivariate Gaussian, (called TGVAE) both for mutation effect prediction and variants design. We show that TDVAE is robust to parameters settings and can have significant performance improvements compared to TGVAE, albeit this varies depending on the protein. Importantly, when used TGVAE to design AGAL variants, it failed to identify mutational hotspots and most of the beneficial mutations.

****Minor points of feedback****

Line 116 – 119 is confusing, redundant, and not true. By looking at figure 1a, DeepSequence outperforms TDVAE in at least 8 datasets, and similarly outperformed TDVAE (low mem) in at least 8 datasets.

We thank the reviewer for the comments, and we have now substantially revised the manuscript to clarify each statement to minimize ambiguity.

Figure 1a – In practice, one would compare the DMS against the mean over the 5 replicates. If you did that where would the correlation fall?

Figure 1b-d The protein names should be included in the figure.

We have tried different configurations to account for this, but it is not estetically pleasing and leads to overly crowded picture. We have reported the relevant protein names in the caption.

Why does the low mem model outcompete the full model?

We thank the reviewer for this comment, and we have now discussed it in the manuscript. Our hypothesis is that training on smaller batches allows to somehow regularize the training process in this case, somewhat similar to masked training.

Line 146 - What is the meaning of robust in this context?

We thank the reviewer for the comments, and we have now substantially revised the manuscript to clarify each statement to minimize ambiguity.

Line 165 – 169 – confusing

We thank the reviewer for the comments, and we have now substantially revised the manuscript to clarify each statement to minimize ambiguity.

Reviewer #1 (Remarks to the Author):

The authors did not provide the source code and the implementation details of their method TDVAE. It is necessary to provide the source code for reproducibility in methodological research.

Reviewer #1 (Remarks on code availability):

The authors did not provide the source code.

Reviewer #3 (Remarks to the Author):

This work provides a novel VAE-based generative model called TDVAE with potential applications for variant effect prediction and functional protein generation. The main methodological novelty includes the use of a Dirichlet prior instead of Gaussian prior on the latent code and using an autoregressive model for generation. As requested by the editor, here are our comments for the authors' response to reviewer 2. Before this manuscript can be published in Nature Communications, these points should be addressed. Moreover, in this reviewer's opinion, the code and model weights should be publicly released before acceptance of the manuscript.

Major points

1. The reviewer is satisfied with the author's experimental results of comparing TDVAE with TGVAE. However, the justification for the use of Dirichlet distribution because protein sequences are categorical data is not necessarily an argument that stands, since the prior is applied to the latent space rather than directly modeling the probabilities of the multinomial distribution. In addition, the latent space samples are also not treated as probabilities of multinomial distribution. Dirichlet prior might still be preferable in some cases, but not necessarily for the reason that the authors provided.
2. The authors should conduct a more comprehensive comparison / benchmarking including comparison with previous methods for variant effect prediction [1,2].
3. The reviewer agrees with the author's explanation and justification. Additionally, these questions need to be addressed to gain a comprehensive and strengthened interpretation for the model and the results:
 - i. How similar are generated sequences to real sequences in the training data for the AGAL and how diverse are the generated sequence? The authors should provide the generate sequences for a detailed evaluation of generative ability for the reviewers.
 - ii. Coevolution signal is an important metric to consider when evaluating the biochemical and functional properties for protein family sequences. Does the model capture any signal of coevolution when trained on DMS data?
4. Satisfied.

Minor points.

1. Satisfied.
 2. The authors did not address this issue.
 3. Satisfied.
 4. The author should give a more detailed explanation on why training on smaller batches could serve as a way of regularization.
 5. The word robust is still abused throughout the manuscript without further justification. For example, see line 364.
 6. Satisfied.
- Below is another minor point:
1. Line 379: typo: them them.

References:

- [1] Riesselman, Adam J., John B. Ingraham, and Debora S. Marks. 2018. "Deep Generative Models of Genetic Variation Capture the Effects of Mutations." *Nature Methods* 15 (10): 816–22.
- [2] Notin, Pascal, Mafalda Dias, Jonathan Frazer, Javier Marchena Hurtado, Aidan N. Gomez, Debora Marks, and Yarin Gal. 2022. "Tranception: Protein Fitness Prediction with Autoregressive Transformers and Inference-Time Retrieval." In *Proceedings of the 39th International Conference on Machine Learning*, 16990–17. PMLR.

Reviewer #3 (Remarks on code availability):

I dont

RESPONSE TO REVIEWERS

REVIEWER #1

The authors did not provide the source code and the implementation details of their method TDVAE. It is necessary to provide the source code for reproducibility in methodological research.

The authors would like to confirm that, upon acceptance, we will make the software freely available to the academic community at the following url: <https://licensing.edinburgh-innovations.ed.ac.uk>.

REVIEWER #3

Major points

1. The reviewer is satisfied with the author's experimental results of comparing TDVAE with TGVAE. However, the justification for the use of Dirichlet distribution because protein sequences are categorical data is not necessarily an argument that stands, since the prior is applied to the latent space rather than directly modeling the probabilities of the multinomial distribution. In addition, the latent space samples are also not treated as probabilities of multinomial distribution. Dirichlet prior might still be preferable in some cases, but not necessarily for the reason that the authors provided.

We thank the reviewer for the comment, and we have further revised our statements to avoid any confusion; regarding the benefit of Dirichlet latent modelling, we reasoned that this is likely to the inherent properties of this distribution to capture multimodalities (see Discussion).

2. The authors should conduct a more comprehensive comparison / benchmarking including comparison with previous methods for variant effect prediction [1,2].

We appreciate reviewers' feedback, but we do think that adding even more benchmarking is significantly impacting the timeliness of our work for few reasons:

1. It adds very little in terms of validation, since we have already shown on a standard, widely used dataset that we are at the very least on par. Moreover, we toned down the initial claims and emphasized how this part of our work serves as validation of the learning capabilities of our model and its efficiency, rather than claiming to be the best mutation effect predictor.
2. We feel like adding more dataset/models is beyond the scope of this paper, since the vast majority (if not all) the code referenced requires substantial work and configuration before carrying out experiments. Although valuable, we think this work would be more suitable for a benchmarking manuscript.

In consultation with the editor, it was decided not to do the additional benchmarking requested.

3. The reviewer agrees with the author's explanation and justification. Additionally, these questions need to be addressed to gain a comprehensive and strengthened interpretation for the model and the results:

- i. How similar are generated sequences to real sequences in the training data for the AGAL and how diverse are the generated sequence? The authors should provide the generate sequences for a detailed evaluation of generative ability for the reviewers.*
- ii. Coevolution signal is an important metric to consider when evaluating the biochemical and functional properties for protein family sequences. Does the model capture any signal of coevolution when trained on DMS data?*

We thank the reviewer for the feedback. Regarding point a) we discussed the generative ability for AGAL from line 210 to 280, including a full sequence analysis as depicted in fig 4.

It is important to note that we generated a library by seeding the model with the wildtype protein in order to test whether it could generate putative functional variants for lab testing. Optimising metric such as diversity, it is easily achievable, e.g. by sampling from the prior, but has little practical application. We made this point clear from line 186 to 209. On point b) we did not measure coevolution, as most datasets include single point mutations, so they are not useful in this context.

Minor points.

1. *Satisfied.*

2. *The authors did not address this issue.*

We did report and compare models both using the best absolute performance and the average over 5 runs. The results are reported in lines 122-134.

3. *Satisfied.*

4. *The author should give a more detailed explanation on why training on smaller batches could serve as a way of regularization.*

We do hypothesize that, for some datasets where fitness variation is limited i.e. little variation between mutants, having higher variance in gradient estimates might help overcome converge to a local minimum. We think this hypothesis will need further investigation that is outside the scope of the paper, possibly using different architectures for encoders and decoders.

5. *The word robust is still abused throughout the manuscript without further justification. For example, see line 364.*

Removed.

6. *Satisfied.*

Below is another minor point:

7. *Line 379: typo: them them.*

Fixed.

References:

[1] Riesselman, Adam J., John B. Ingraham, and Debora S. Marks. 2018. "Deep Generative Models of Genetic Variation Capture the Effects of Mutations." *Nature Methods* 15 (10): 816–22.

[2] Notin, Pascal, Mafalda Dias, Jonathan Frazer, Javier Marchena Hurtado, Aidan N. Gomez, Debora Marks, and Yarin Gal. 2022. "Tranception: Protein Fitness Prediction with Autoregressive Transformers and Inference-Time Retrieval." In *Proceedings of the 39th International Conference on Machine Learning*, 16990–17. PMLR.

Reviewer #3 (Remarks on code availability):

The authors would like to confirm that, upon acceptance, we will make the software freely available to the academic community at the following url: <https://licensing.edinburgh-innovations.ed.ac.uk>.

Reviewer #3 (Remarks to the Author):

The authors have addressed my points. The final manuscript should include a link to the website URL provides the code.

RESPONSE TO REVIEWERS

REVIEWER #3

The authors have addressed my points. The final manuscript should include a link to the website URL provides the code.

The authors thank the reviewer for the useful comments, and we have now added the URL of the source code in the "Code Availability" section.